# Leptin Receptor Mediates Bmal1 Regulation of Estrogen Synthesis in Granulosa Cells

**DOI:** 10.3390/ani9110899

**Published:** 2019-11-01

**Authors:** Guiyan Chu, Guangjun Ma, Jingchun Sun, Youbo Zhu, Aoqi Xiang, Gongshe Yang, Shiduo Sun

**Affiliations:** 1Key Laboratory of Animal Genetics, Breeding and Reproduction of Shaanxi Province, College of Animal Science and Technology, Northwest A&F University, Yangling 712100, China; guiyanchu@nwafu.edu.cn (G.C.); m1580009184@163.com (G.M.); SunJC7497@nwafu.edu.cn (J.S.); zhuyoubo_2015@nwafu.edu.cn (Y.Z.); aoqixiang0305@126.com (A.X.); gsyang@nwafu.edu.cn (G.Y.); 2Laboratory of Animal Fat Deposition and Muscle Development, College of Animal Science and Technology, Northwest A&F University, Yangling 712100, China

**Keywords:** circadian clock, Bmal1, leptin receptor, granulosa cells, estrogen

## Abstract

**Simple Summary:**

There is increased interest in determining the effect of the biological clock system on reproduction, but how this biological system affects mammalian fertility and the regulation by clock genes on key genes of reproduction is poorly understood. This study examined the function of Leptin on reproduction through interaction with the Leptin receptor (Lepr) and the regulation of the key clock gene brain and muscle ARNT-like 1 (Bmal1) on Lepr. The results suggested that estrogen (E_2_) synthesis is regulated by Bmal1 through the Leptin–Lepr pathway as part of the regulatory mechanism of the circadian system on the fertility of female mammals.

**Abstract:**

Chronobiology affects female fertility in mammals. Lepr is required for leptin regulation of female reproduction. The presence of E-box elements in the *Lepr* promoter that are recognized and bound by clock genes to initiate gene transcription suggested that circadian systems might regulate fertility through *Lepr*. However, it is unclear whether Bmal1, a key oscillator controlling other clock genes, is involved in leptin regulation in hormone synthesis through Lepr. In this study, serum estradiol (E_2_) concentration and the expressions of *Bmal1*, *Lepr*, *Cyp19a1*, and *Cyp11a1* genes were found to display well-synchronized circadian rhythms. Knockdown of *Bmal1* significantly reduced expression levels of *Lepr*, *Fshr*, and *Cyp19a1* genes; protein production of Bmal1, Lepr, and Cyp19a1; and the E_2_ concentration in granulosa cells. Knockdown of *Lepr* reduced the expression levels of *Cyp19a1* and *Cyp11a1* genes and Cyp19a1 protein, and also reduced E_2_ concentration. Addition of leptin affected the expression of *Cyp19a1*, *Cyp11a1*, and *Fshr* genes. *Bmal1* deficiency counteracted leptin-stimulated upregulation of the genes encoding E_2_ synthesis in granulosa cells. These results demonstrated that Bmal1 participates in the process by which leptin acts on *Lepr* to regulate E_2_ synthesis.

## 1. Introduction

Female fertility depends on a functional circadian system, which has an intimate association with the reproduction processes [1]. At the molecular level, the circadian clock is composed of a network of transcription-translation feedback loops. Heterodimers of Clock (circadian locomotor output cycles kaput) and Bmal1 bind to E-box elements on the genome to activate the transcription of a number of clock control genes [2]. These clock control genes are expressed and coordinate with rhythmic physiological functions in cells and tissues [3]. Bmal1 is the key gene of this oscillator, and its deficiency results in severe clock disruption, which has been associated with decreased fertility [4,5,6,7]. The extent of illumination as well as exposure to light at night can cause circadian clock dysfunction, which increases rates of menstrual disruption, odds ratio, and subfertility in women [8]. However, the details of the regulation of steroidogenesis by the circadian clock have not been determined.

Leptin, encoded by the *obese* gene [9], is a metabolic hormone [10]. Leptin is predominantly produced by adipose tissues, with some amount produced by other tissues, such as the stomach, muscle, ovary, and placenta [11,12]. Via its specific receptors, leptin acts to regulate food intake and energy metabolism [13]. In addition, leptin is essential for fertility due to its ability to affect the functions of the hypothalamus, pituitary, and reproductive organs [10]. *Leptin* mutants (*ob*/*ob*) mice are infertile, but the administration of exogenous leptin raises FSH and LH concentrations, increases gonadal weight, and restores fertility [14,15]. Lepr is encoded by the *diabetes* gene [16] and belongs to the class-1 cytokine receptor superfamily [17]. Transcription and translation of Lepr occur in different tissues, including hypothalamus, pituitary, ovary, and adrenal glands [18]. The activation of specific membrane-associated receptors is required for leptin function [13,19]. Crucially, human chorionic gonadotropin (HCG) stimulates Lepr expression in granulosa cells [18] and Lepr null mice are infertile [20]. In some mammal granulosa cells, leptin can promote cell proliferation, impair cell apoptosis, and stimulate steroidogenesis through the MAPK and PKA pathways in a dose-dependent manner [21]. In oocytes, leptin enhances nuclear and cytoplasmic maturation [9]. Mice treated with leptin antagonists exhibit significantly reduced ovulation rates [18].

Leptin works through Lepr in the ovary [22]. Measurements of mRNA and protein of Leptin and ovarian cells suggest that leptin directly regulates ovarian cell functions. Although the *Lepr* promoter includes potential binding sites for Bmal1/Clock heterodimers, whether and how Bmal1 participates in the leptin-regulated synthesis of E_2_ by affecting the expression of Lepr has been determined. Therefore, in this study, we focused on the role of Bmal1 in the process of leptin-regulated E_2_ synthesis.

## 2. Materials and Methods

### 2.1. Animals, Welfare Assurance, and Management

The use of animals and the experimental protocol were approved by the Northwest Agriculture and Forestry University Animal Research Ethics Committee (Yangling, Shaanxi, China).

Three-week old, Kunming background female mice were purchased from the Laboratory Animal Center of the Fourth Military Medical University (Xi’an, China). Rodents were housed at 2–5 per cage, fed ad libitum a standard laboratory chow diet, and had free access to fresh water. The animal room was maintained at a constant temperature of 25 ± 1 °C and humidity at 55% ± 5% with 12 h light/12 h dark cycles (light, ZT0–12; dark, ZT12–24; ZT = zeitgeber time). 

### 2.2. Granulosa Cell Culture and Treatment

Granulosa cells were harvested as described previously [23] with some minor changes. Briefly, mice (21–23 d of age) were injected subcutaneously with 1 mg of diethylstilbestrol (DES; Sigma-Aldrich, Inc., St. Louis, MO, USA) for 3 d. Granulosa cells were harvested from ovary follicles on day 4 at ZT1. Ovaries were incubated with DMEM/F12 containing 6 mM EGTA for 20 min and then incubated in DMEM/F12 medium containing 0.5 M sucrose at 37 °C for 15 min before granulosa cells were collected. After washing with PBS, follicles were punctured with a 27-gauge needle in DMEM/F12 medium and then passed through a 70 and then 200-mesh sieve. Granulosa cells were washed by centrifugation at 1000 rpm for 10 min and seeded in culture dishes. The cells were cultured in DMEM/F12 medium supplemented with 10% FBS (Gibco Laboratories, Gaithersburg, MD, USA) at 37 °C in 5% CO_2_ for 24 h and then the medium was substituted with serum-free medium containing 0.3% Albumin Bovine V (Beijing Solarbio Science and Technology Co., Ltd., Beijing, China) for an additional 12 h. For leptin treatment, cells were cultured in fresh media containing 0, 0.1, 0.5, 1, 5, and 10 ng/mL of recombinant mouse leptin (R&D Systems, Inc., Minneapolis, MN, USA) for 24 h.

### 2.3. Transfection with siRNA 

The siRNA technique was used to assess the effects of *Bmal1* and *Lepr* on the oscillation-related genes in granulosa cells to determine *Bmal1* and *Lepr* function for E_2_ synthesis. The siRNA sequences of *Bmal1*, *Lepr*, and a scrambled negative control (NC) were designed and synthesized by Shanghai GenePharma Co., Ltd. (Shanghai, China). The sequences of siRNA used in this study are as follows:

*Bmal1*: 5′-CCUCCACAAUCAGUGACUUTT-3′ (sense) and 5′-AAGUCACUGAUUGUGGAGGTT-3′ (antisense),

*Lepr*: 5′-GCUGGAGUCCCAAACAAUATT-3′ (sense) and 5′-UAUUGUUUGGGACUCCAGCTT-3′ (antisense), and

NC: 5′-UUCUCCGAACGUGUCACGUTT-3′ (sense) and 5′-ACGUGACACGUUCGGAGAATT-3′ (antisense). 

The transfection protocol was performed according to the manufacturer’s guidelines. When the cell density reached 50%–60%, the cell culture medium was replaced with new medium without serum. The siRNA and NC were diluted into Opti-MEM (Gibco Laboratories) and compounded with X-tremeGENE siRNA transfection reagent (Roche Applied Science, Indianapolis, IN, USA) and delivered into the cells. After exposure for 24 h, the medium was replaced by DMEM/F12 medium containing 0.3% bovine albumin fraction V (Beijing Solarbio Science and Technology Co., Ltd.) and incubated for 24 h. 

### 2.4. RNA Extraction and Quantitative Real-Time PCR Measurement

Total RNA samples were isolated from ovaries and cultured cells using RNAiso Plus (TaKaRa Bio Inc., Shiga, Japan) according to the manufacturer’s protocol [24]. After extraction, 500 ng of total RNA was used to produce cDNA with a Prime Scrip RT Kit (TaKaRa Bio Inc.). Specific primers were designed using Primer 5.0 software and GAPDH used as a housekeeping gene for comparison. Real-time PCR was carried out with IQ5 system using SYBR Green in a mixture system including 5 μL of SYBR (Vazyme Biotech Co., Ltd., Nanjing, China), 0.5 μL of forward primers, 0.5 μL of reverse primers, 1 μL of template cDNA, and 3 μL of ddH_2_O. Data were calculated with the 2-∆∆t method. Primer sequences are listed in Table 1 and Table 2.

### 2.5. Western Blotting

Cellular proteins were extracted from ovaries and cultured granulosa cells with RIPA (Applygen Technologies Inc., Beijing, China) supplemented with 1 mM PMSF serine protease inhibitor. The protein content was measured using a BCA Protein Assay Kit according to the manufacturer’s guidelines (CWBIO, Beijing, China). Western blotting was conducted as described in a previous report [24]. A total of 20 micrograms of protein was electrophoresed on a 12% SDS-polyacrylamide gel and transferred to PVDF membranes. The levels of the Bmal1 and other proteins involved in Lepr function and E_2_ synthesis were measured using specific antibodies. Antibodies against Bmal1 (1/500; sc-365645, Santa Cruz Biotechnology, Inc., Santa Cruz, CA, USA), Lepr (1/500; ab5593, Abcam, Inc., Cambridge, MA, USA), Fshr (1/1000; ab75200, Abcam, Inc.), Cyp19a1 (1/1000; ab124776, Abcam, Inc.), and Cyp11a1 (1/500; #14217, CST, Inc., Danvers, MA, USA) were used to detect the levels of these proteins in granulosa cells. 

### 2.6. Estradiol Assay

The E_2_ concentrations in the mouse serum samples and cell culture medium were detected using an ELISA assay (Nanjing Jiancheng Bio-Engineering Institute Co., Ltd., Jiancheng, China) in accordance with the manufacturer’s protocol.

### 2.7. Statistical Analysis

Statistical analysis was performed using Graphpad Prim6 [25]. Student’s *t*-test and one-way ANOVA were used to analyze paired groups and more than two groups. Data are presented as mean ± SE. Significances were indicated as *, *p* < 0.05 and **, *p* < 0.01. Relative gene expression was analyzed by the 2-∆∆ ct method. 

## 3. Results

### 3.1. Leptin Function in Steroidogenesis-Related Genes

To investigate the action of leptin on the expression of steroidogenesis genes, varied leptin doses (0, 0.1, 0.5, 1, 5, and 10 ng/mL) were added into cell culture media for 24 h and the expression levels of the E_2_ synthesis-related genes of *Cyp19a1*, *Cyp11a1*, and *Fshr* were determined. The observed changes of gene expression with changes in the leptin concentration series were different for different genes (Figure 1). For the *Cyp19a1* gene, the maximum expression appeared at 0.5-ng/mL leptin (*p* < 0.05), and the minimum expression was at 10-ng/mL leptin (*p* < 0.05, Figure 1A). The expression of *Cyp11a1* was higher (*p* < 0.05, Figure 1B) at 0.1, 0.5, and 1-ng/mL leptin concentrations compared to the levels at the other leptin concentrations. *Fshr* expression was similar at 0, 0.1, 0.5, and 1 ng/mL leptin but was decreased at 5 and 10 ng/mL leptin (*p* < 0.05, Figure 1C). 

### 3.2. Lepr is Necessary for E_2_ Synthesis

To determine if the observed biological activity of leptin on steroidogenesis required interaction with *Lepr*, the expression levels of *Lepr*, *Cyp19a1*, *Cyp11a1*, and *Fshr* were measured before or after the silencing of *Lepr* by siRNA in the presence or absence of leptin. *Lepr* siRNA treatment suppressed the leptin-stimulated upregulation of *Cyp19a1*, *Cyp11a1*, and *Fshr* (*p* < 0.05, Figure 2), as evident by comparison to cells without *Lepr* siRNA treatment in the presence of leptin. The results indicated that *Lepr* siRNA treatment weakened the leptin contribution to regulation of E_2_ synthesis-related genes.

Granulosa cells were treated with *Lepr* siRNA to further explore the effects of Lepr on E_2_ synthesis. With the disturbance of *Lepr* and Cyp19a1 were decreased at both the RNA and the protein level. Cyp11a1 was inhibited by *Lepr* siRNA at the RNA level but not at the protein level. Since the RNA level of *Fshr* did not change significantly, its protein concentrations were not detected (Figure 3A–C). The E_2_ concentration in the medium was reduced significantly by si-Lepr (*p* < 0.01, Figure 3D).

### 3.3. E_2_ Concentration in Serum and Lepr Expression in Ovaries Presented Evident Rhythm 

Bioinformatics analysis identified eleven E-box like elements in the 1500 bp upstream of the *Lepr* promoter (Figure 4). Therefore, we next asked if *Lepr* might be regulated by the clock system. To determine whether *Lepr* transcription and E_2_ production were affected by the circadian system, the E_2_ concentrations and mRNA concentrations of ovarian genes (*Bmal1*, *Lepr*, *Fshr*, *Cyp19a1*, and *Cyp11a1*) were tested.

Eight-week-old female mice living in 12 h light/12 h dark cycles were euthanized at ZT 0, 4, 8, 12, 16, 20, and 24 (ZT 0 is 8 am; *n* = 6 per group). The rhythmicity change of *Bmal1* indicated effective function of the circadian system in ovaries of Kunming background female mice (Figure 5A). The *Lepr* and E_2_ mRNA levels showed robust rhythms and both peaked during the dark phase at ZT16 (Figure 5B,F), which suggested that *Lepr* was sensitive to light rhythm and might act in the ovary at ZT16.

The *Bmal1* siRNA was used to interfere with *Bmal1* function in granulosa cells to confirm whether the E_2_ concentration was affected by *Bmal1*. The results showed decreased *Lepr* and *Cyp19a1* RNA and protein levels in the presence of the *Bmal1* siRNA. Fshr was inhibited by *Bmal1* siRNA at the RNA level but not at the protein level. Since *Cyp11a1* did not change significantly with the *Bmal1* siRNA, its protein concentration was not detected. The results showed that transcription of the *Clock* gene was not affected by *Bmal1* interference, which was consistent with the previous report (Figure 6A–C). Consistently, the E_2_ level was significantly suppressed by the *Bmal1* siRNA (Figure 6D). 

### 3.4. Bmal1 siRNA Repressed Leptin-Induced E_2_ Synthesis-Related Genes

We next asked if leptin is required for the observed effect of silencing *Bmal1*. To determine if Bmal1 is involved in the hormone synthesis process that is promoted by leptin, granulosa cells were cultured with *Bmal1* siRNA for 24 h and then treated with 0.5-ng/mL leptin for 24 h. The results showed that *Bmal1* siRNA prevented the leptin-promoted upregulation of Lepr, Cyp19a1, Cyp11a1, and Fshr (*p* < 0.05, Figure 7A–E).

## 4. Discussion

E_2_ concentration and E_2_ synthesis-related gene expression were altered by adding leptin to cell media, indicating that leptin affected E_2_ synthesis. The down regulation of E_2_ synthesis-related gene expression caused by *Lepr* siRNA indicated the involvement of *Lepr* in E_2_ synthesis regulation in granulosa cells. This observation was consistent with the effect of leptin on granulosa cells, as previously reported. For example, in cultured rabbit granulosa cells, addition of 10 ng/mL leptin promoted proliferation, reduced apoptosis, and increased E_2_ generation via excitation of the MAPK and PKA signaling pathways [16]. The present results indicated that *Fshr* down regulation with 10 ng/mL leptin was consistent with the previous report that FSH stimulation of steroidogenes is inhibited by high doses of exogenous leptin [26].

The expression patterns of *Bmal1*, *Lepr*, *Fshr*, *Cyp19a1*, and *Cyp11a1* genes in ovarian cells were found to exhibit circadian rhythms within the 12 h light/12 h dark cycle, as did the serum E_2_ concentration. All patterns were well-synchronized, and peaked at ZT16 (dark period). Thus, the findings suggested the involvement of Lepr in E_2_ synthesis and that this physiological process might be regulated by the circadian system.

Expression of ovarian genes and E_2_ concentrations in cultured granulosa cells were reduced with Bmal1 deficiency, which suggested that Bmal1 was essential to maintain the functions of genes for ovarian steroidogenesis. These results were consistent with a previous report that specific knockout of *Bmal1* in ovarian steroidogenesis cells resulted in abnormalities of mice estrus cycles [27]. Wang proposed that *Bmal1* deficiency increased cell apoptosis and inhibited steroidogenesis in porcine granulosa cells [24]. *Lepr* transcription in ovarian cells was regulated by Bmal1 and displayed rhythmicity because Bmal1 deficiency impaired Lepr expression and counteracted leptin-promoted expression of the E_2_ synthesis genes in granulosa cells. Both Lepr and E_2_ synthesis-related genes were repressed when *Bmal1* was inhibited by siRNA even in the presence of leptin, but this repression did not occur when *Bmal1* was normally expressed. Thus, the results indicated that Bmal1 is necessary for leptin action in the process of E_2_ synthesis. 

## 5. Conclusions

In conclusion, the results indicate that Lepr was involved in the stimulation of E_2_ synthesis in granulosa cells by leptin, and this process might be regulated by the biological clock gene Bmal1 (Figure 8). There were eleven circadian binding sites in the promoter of lepr, but whether Bmal1 directly regulates *Lepr* and the details of the regulatory mechanism must be further verified.

## Figures and Tables

**Figure 1 animals-09-00899-f001:**
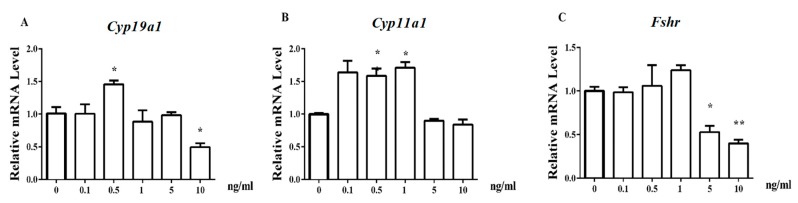
Leptin regulates the expression of E_2_ synthesis-related genes of *Cyp19a1*, *Cyp11a1*, and *Fshr* in a dose-dependent manner in granulosa cells after exposure to leptin (0, 0.1, 0.5, 1, 5, and 10 ng/mL) for 24 h. (**A**–**C**): relative mRNA expression of *Cyp19a1*, *Cyp11a1*, and *Fshr*, respectively. Statistics are showed as mean ± SEM (*n* = 3 per group). Significances are expressed as follows: * *p* < 0.05, ** *p* < 0.01.

**Figure 2 animals-09-00899-f002:**
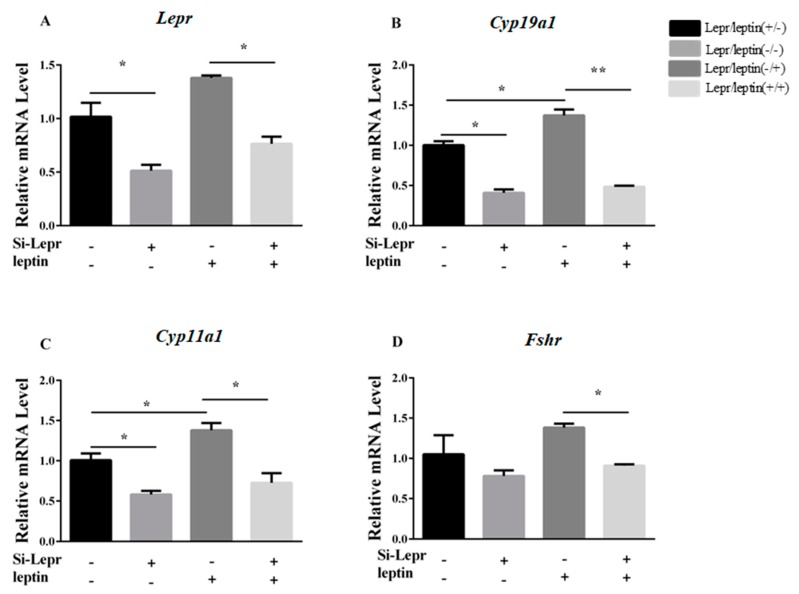
The interference of *Lepr* impaired the action of leptin on synthesis genes. After treatment with Lepr siRNA for 24 h, granulosa cells were cultured with or without leptin (0.5 ng/mL) for 24 h. The transcription levels of *Lepr*, *Cyp19a1*, and *Cyp11a1* were determined by qRT-PCR. (**A**–**D**): relative mRNA expression of *Lepr*, *Cyp19a1*, *Cyp11a1*, and *Fshr*. Statistics are presented as mean values ± SEM (*n* = 3 per group). Significance values are expressed as follows: * *p* < 0.05, ** *p* < 0.01 (si-Lepr: Lepr siRNA treatment group, NC: negative control group).

**Figure 3 animals-09-00899-f003:**
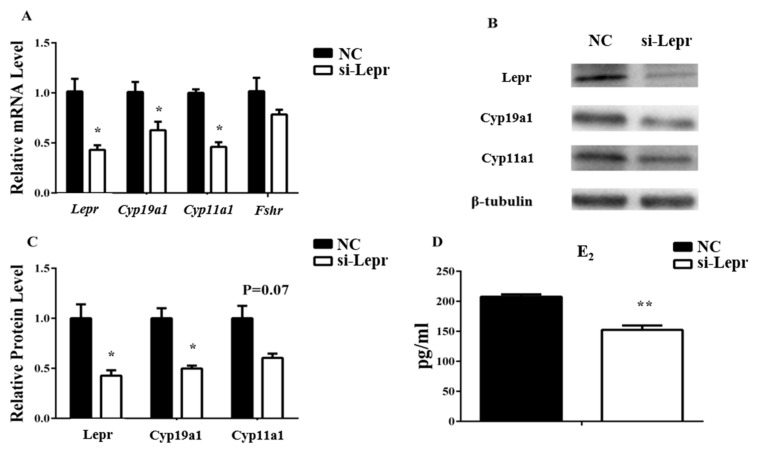
*Lepr* siRNA treatment reduces E_2_ production in granulosa cells. The transcription levels of ovarian genes were ascertained by qRT-PCR. The protein levels were analyzed by western blot. The E_2_ concentrations in the cell medium were measured by ELISA. (**A**): relative mRNA expression of *Lepr*, *Cyp19a1*, and *Cyp11a1*. (**B**,**C**): relative protein expression of Lepr, Cyp19a1, and Cyp11a1. (**D**): E_2_ concentration in cell medium. Results are presented as mean values ± SEM (*n* = 3 per group). The significance of the results is expressed as follows: * *p* < 0.05, ** *p* < 0.01 (si-Lepr: *Lepr* siRNA treatment group, NC: negative control group).

**Figure 4 animals-09-00899-f004:**
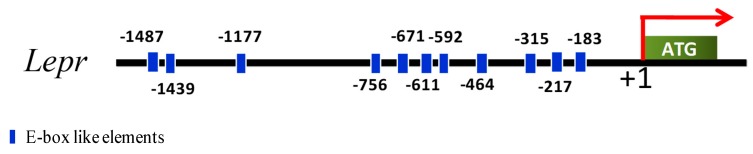
Schematic representation of the porcine *Lepr* gene promoter. Eleven E-box like elements are located upstream of the promoter.

**Figure 5 animals-09-00899-f005:**
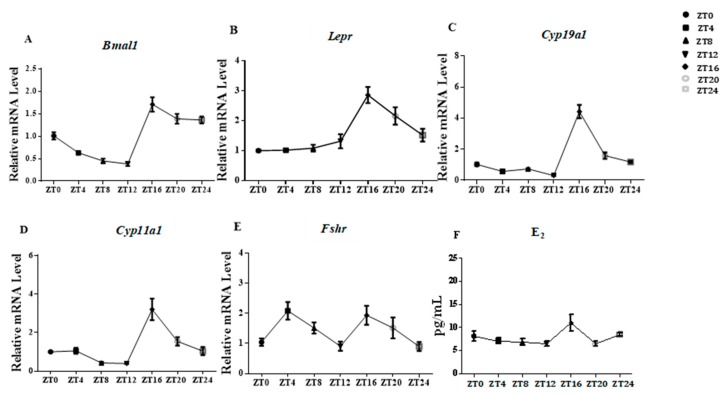
The rhythm of E_2_ concentration in serum and related gene expression in ovaries. Samples were from eight-week-old female mice subjected to light/dark cycles. The transcription levels of ovarian genes were ascertained by qRT-PCR at ZT0, ZT4, ZT8, ZT12, ZT16, ZT20, and ZT24. The E_2_ concentrations in blood sera were measured by ELISA at ZT0, ZT4, ZT8, ZT12, ZT16, ZT20, and ZT24. (**A**–**E**): the relative mRNA expression levels of ovarian genes. (**F**): E_2_ concentration in blood serum. Statistics are showed as mean values ± SEM (*n* = 6; ZT: zeitgeber time).

**Figure 6 animals-09-00899-f006:**
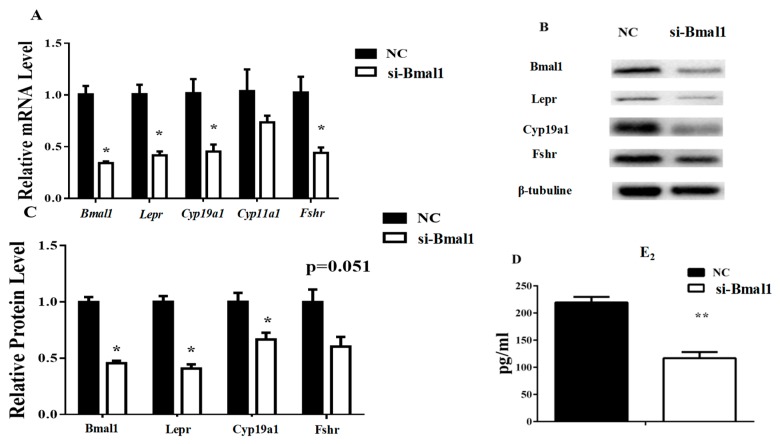
Silencing of *Bmal1* impairs the expression of Lepr and the secretion of E_2_ in granulosa cells. The transcription levels of ovarian genes were measured by qRT-PCR and the protein levels were analyzed by western blot. The E_2_ concentration in cell medium was measured by ELISA. (**A**): relative mRNA expression of clock and ovarian genes. (**B**,**C**): relative protein expression of Bmal1, Lepr, Fshr, and Cyp19a1. (**D**): E_2_ concentration in cell medium. Statistics are showed as mean ± SEM (*n* = 3 per group). Significances are expressed as follows: * *p* < 0.05, ** *p* < 0.01 (si-Bmal1: *Bmal1* siRNA treatment group, NC: negative control group).

**Figure 7 animals-09-00899-f007:**
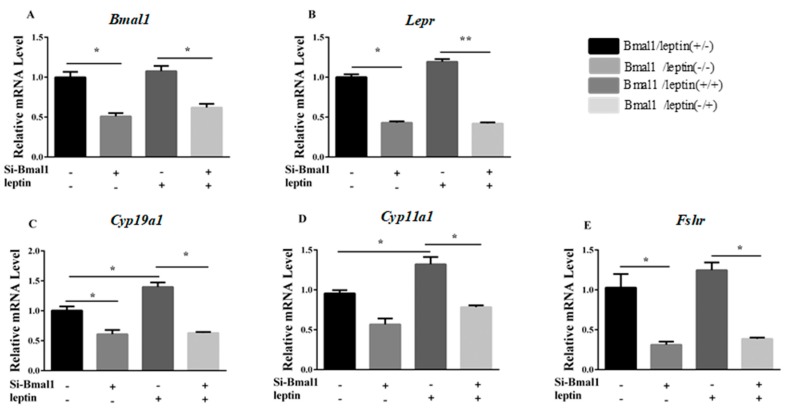
Silencing of *Bmal1* limited the action of leptin on synthesis genes. After treatment with *Bmal1* siRNA for 24 h, granulosa cells were cultured with or without leptin (0.5 ng/mL) for 24 h. (**A**–**E**): relative mRNA expression of *Bmal1*, *Lepr*, *Cyp19a1*, *Cyp11a1*, and *Fshr* were determined by qRT-PCR. Statistics are shown as mean ± SEM (*n* = 3 per group). Significance values are expressed as follows: * *p* < 0.05, ** *p* < 0.01 (si-Bmal1: *Bmal1* siRNA treatment group, NC: negative control group).

**Figure 8 animals-09-00899-f008:**
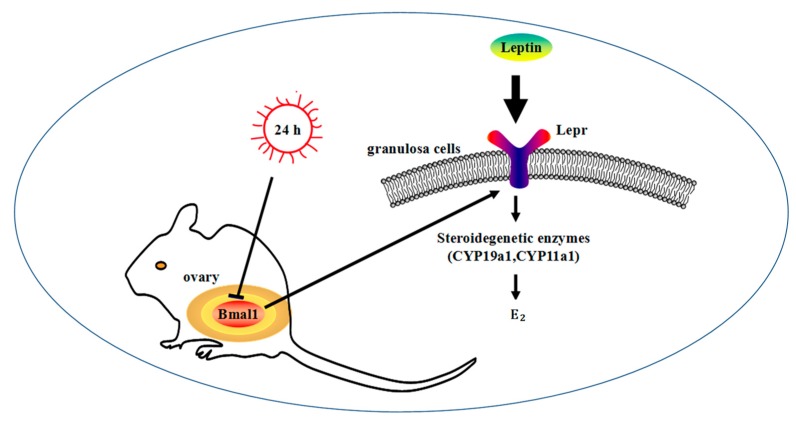
*Lepr* was regulated by Bmal1 in the leptin-regulated process of E_2_ synthesis. *Lepr* and E_2_ synthesis-related genes are regulated by Bmal1.

**Table 1 animals-09-00899-t001:** Primers of mice genes.

Gene	Accession No.	Primer Sequence 5′-3′	Length	Tm/°C
*Bmal1*	NM_001357070.1	F: ACAGTCAGATTGAAAAGAGGCG R: GCCATCCTTAGCACGGTGAG	124	60
*Lepr*	NM_010704.2	F: ACCTGGCATATCCAATCTCTCC R: TTCAAAGCCGAGGCATTGTTT	115	60
*Cyp11a1*	NM_001346787.1	F: GGGCAGTTTGGAGTCAGTTTAC R: TTTAGGACGATTCGGTCTTTCTT	186	60
*Cyp19a1*	NM_001348171.1	F: AACCCCATGCAGTATAATGTCAC R: AGGACCTGGTATTGAAGACGAG	132	60
*Fshr*	NM_013523.3	F: TGCTCTAACAGGGTCTTCCTC R: TCTCAGTTCAATGGCGTTCCG	84	60
*Gapdh*	NM_017321385.1	F: TGCTGAGTATGTCGTGGAGTCT R: ATGCATTGCTGACAATCTTGAG	179	60

*Bmal1*: brain and muscle ARNT-like 1; *Lepr*: leptin receptor; *Cyp19a1*: cytochrome P450 family 19 subfamily A member 1; *Cyp11a1*: cytochrome P450 11 subfamily A member 1; *Gapdh*: glyceraldehyde 3-phosphate dehydrogenase; F: forward; R: reverse.

**Table 2 animals-09-00899-t002:** qRT-PCR primers of rats genes.

Gene	Accession No.	Primer Sequence 5′-3′	Length	Tm/°C
*Bmal1*	NM_024362.2	F: ACAGTCAGATTGAAAAGAGGCG R: GCCATCCTTAGCACGGTGAG	124	60
*Lepr*	NM_012596.1	F: CCCACAATGGGACATGGTCA R: GCACCGATGGAATTGATGGC	109	60
*Cyp11a1*	NM_017286.3	F: CAGACGCATCAAGCAGCAAA R: GGTCCACGATCTCCTCCAAC	134	60
*Cyp19a1*	NM_017085.2	F: AGAGACGTGGAGACCTGACA R: CCTCCGGATACTCTGCGATG	126	60
*Fshr*	NM_199237.1	F: ATTCTTGGGCACGGGATCTG R: CGGTCGGAATCTCTGTCACC	98	60
*Gapdh*	NM_017008.4	F: AAGGTCGGTGTGAACGGATT R: CTTTGTCACAAGAGAAGGCAGC	70	60

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
