# Peer review of "Leptin Receptor Mediates Bmal1 Regulation of Estrogen Synthesis in Granulosa Cells"

_animals, 2019, doi:10.3390/ani9110899_

Round 1
Reviewer 1 Report
This manuscript by Dr. Guiyan Chu et al., (Shiduo Sun laboratory) evaluates circadian gene expression and ultimately, estrogen (E2) synthesis in what the authors hypothesize occurs in a circadian machinery controlled fashion. The article is interesting and fits within existing literature. There are two major problems with the manuscript in its current form that reduce enthusiasm - and each of these relate to the clarity of the work. Appreciating that English may not be the first language of the authors, the first problem is that there are extraordinarily numerous errors in English usage and grammar. The article needs massive copyediting so that peer reviewers, and eventually, readers, can correctly interpret the work. The second major problem is that the scientific clarity and logic of the work are very difficult to follow. The authors cover introductory background information in a believable way, but the work needs to be re-framed in a clear hypothesis-driven way. By this I mean that the authors need to clearly state their hypotheses, that 1) Lepr might regulate E2 synthesis (by regulating what genes?) and that 2) circadian machinery including Bmal1 might regulate Lepr. The authors need to create a clear logical path using hypotheses supported by published literature so readers can understand exactly WHY the data were collected.
The only specific scientific issue I have with the paper in its current form is that the authors contend that Bmal1 can be placed upstream of Lepr, which can be placed upstream of E2 (Bmal1 -> Lepr -> E2). This is difficult to understand as circadian machinery most often drives the transcription of circadian-responsive genes. The authors show here that Bmal1, Lepr, and E2, along with steroidigenic transcripts all go up concurrently between 12 and 16 hours. If Lepr (and thus E2) are responding to Bmal1 transcriptional activation, there should be a delay - do the authors interpet their data as showing that peak Bmal1 levels are driving the expression of the downstream genes in the next cycle, 24 hours later?
Last, the authors only show genes that DO respond to Bmal1 knockdown; is this because all of these are actual downstream targets of Bmal1, or, is it possibly due to nonspecific knockdown effects or toxicity to cells after siRNA delivery? A scrambled siRNA control should be added, and at least one gene that does not respond by downregulation after Bmal1 knockdown should be added.
Without dealing with all of the above issues, it would be very difficult to clearly place this work in context with existing literature.
Author Response
Response to Reviewer1 Comments
Point1: the first problem is that there are extraordinarily numerous errors in English usage and grammar. The article needs massive copyediting so that peer reviewers, and eventually, readers, can correctly interpret the work.
Response 1: The manuscript was checked and carefully edited by a native English scientific researcher. The certificate is submitted in the system.
Point 2: The second major problem is that the scientific clarity and logic of the work are very difficult to follow.The authors cover introductory background information in a believable way, but the work needs to be re-framed in a clear hypothesis-driven way. By this I mean that the authors need to clearly state their hypotheses, that 1) Lepr might regulate E2 synthesis (by regulating what genes?) and that 2) circadian machinery including Bmal1 might regulate Lepr. The authors need to create a clear logical path using hypotheses supported by published literature so readers can understand exactly WHY the data were collected.
Response 2: I have rewritten the manuscript as you suggested. The logic is that, first of all, we clarify the role of Leptin and Lepr in estrogen synthesis and the relationship between them. And then we found there were eleven E-boxes like elements in the promoter of Lepr so the regulation of the core biological clock gene Bmal1 was introduced. Therefore, E2 and Lepr rhythms were detected. At last, Bmal1 function in E2 synthesis with the presence of Leptin was detected.
Point 3: The only specific scientific issue I have with the paper in its current form is that the authors contend that Bmal1 can be placed upstream of Lepr, which can be placed upstream of E2 (Bmal1 -> Lepr -> E2). This is difficult to understand as circadian machinery most often drives the transcription of circadian-responsive genes. The authors show here that Bmal1, Lepr, and E2, along with steroidigenic transcripts all go up concurrently between 12 and 16 hours. If Lepr (and thus E2) are responding to Bmal1 transcriptional activation, there should be a delay - do the authors interpet their data as showing that peak Bmal1 levels are driving the expression of the downstream genes in the next cycle, 24 hours later?
Response 3: I agree with the viewer if there are only circadian binding E-boxes or E-box like elements in the promoter of Lepr. But there are also other elements in the promoter; therefore, the results showed that Bmal1, Lepr, and E2, along with steroidigenic transcripts all go up concurrently between 12 and 16 hours. In order to explain clearly, we have listed the E-box like elements in the promoter of Lepr.
Point4: Last, the authors only show genes that DO respond to Bmal1 knockdown; is this because all of these are actual downstream targets of Bmal1, or, is it possibly due to nonspecific knockdown effects or toxicity to cells after siRNA delivery? A scrambled siRNA control should be added, and at least one gene that does not respond by downregulation after Bmal1 knockdown should be added.
Response 4: In the experiment with siBmal1, we used a scrambled negative control as CN and the sequence was also listed in the manuscript, so we think the results showed the specific knockdown effects of Bmal1.

Reviewer 2 Report
Comments to animals-591179-peer-review-v1
Major comments
In this paper, authors described the evidences that Lepr mediated Bmal1 regulation of E2 synthesis. Experiments were well-desingned, the methods are reasonable and the narrative flows well. Only the presentation and the interpretation of the data related to Fig. 3 does not seem to be precise. Comments listed below should be carefully addressed.
Minor comments are listed below:
Line 76: “).” at the end of the last sentence should be deleted.
Line 81: “DMEM/12” should be “DMEM/F12”.
Line 86: “FBS” is fetal bovine serum, thus “serum” should be removed.
Line 90: “mg/mL” should be removed.
Line 105: “mem” should be “MEM”.
Line 126: “20 mg” may be “20 micro g ”.
Lines 128-131: “Primary antibodies….. for E2 synthesis” should be grammatically wrong and re-written.
Lines 132-134: This part is also found in lines 94-96.
M&M: There is no description on the sampling procedures of sera and ovaries used for obtaining the results in lines 146-160. Therefore, lines 149-150, “The 8-wee-old….(n=6 per group), should be moved to M&M.
Lines 162-163: “a series of …. and 10 ng/mL)” needs to be re-written.
Lines 162-169: There is no reasonable explanation/discussion on the quadratic response of the expression of cyc19a1, cyp11a1 and fshr genes to the leptin in any sections.
Line 173: Is “RIA” correct? The authors describes “ELISA” in line 137.
Line 174-175: Check these sentences. The figure legends for “C and D” are wrong.
Line 180-181: Leptin did not stimulate Fshr gene expression, thus this sentence should be re-written.
Line 189: “Cyp9a1” should be “Cyp19a1”.
Lines 196-197: A-D are not marked in Fig.3. This reviewer cannot imagine that cyp11a1 mRNA levels did not change by Lepr siRNA treatment from Figure 3. Please carry out the statistical analysis using the data for the potential Fig.3B. The authors stated they did not detect the protein of cyp11a1 but there are statement of cyp11a1 in the potential Fig. 3C and D. This part is so complicated to be understood.
There are possibly other careless and grammatical mistakes through the manuscript. Please check it out, again.
Author Response
Response to Reviewer 2 Comments
Point 1: Line 76: “).” at the end of the last sentence should be deleted.
Response 1: I have deleted this sentence as you suggested.
Point 2: Line 81: “DMEM/12” should be “DMEM/F12”.
Response2: Thank you. I have corrected it.
Point 3: Line 86: “FBS” is fetal bovine serum, thus “serum” should be removed.
Response 3: I have deleted “serum”as you suggested.
Point 4: Line 90: “mg/mL” should be removed.
Response 4: I have removed the first “mg/mL”.
Point 5: Line 105:“mem”should be “MEM”.
Response 5: I have replaced “mem” with“MEM”.
Point 6: Line 126: “20 mg” may be “20 micro g ”.
Response 6: I have replace “20 mg” with “20 micro g”.
Point 7: Lines 128-131: “Primary antibodies….. for E2 synthesis” should be grammatically wrong and re-written.
Response 7: We have written the part with “The levels of the Bmal1 and other proteins involved in Lepr function and E2 synthesis were measured using specific antibodies. Antibodies against Bmal1 (1/500; sc-365645, Santa Cruz Biotechnology, Inc., Santa Cruz, CA, USA), Lepr (1/500; ab5593, Abcam, Inc., Cambridge, MA, USA), Fshr (1/1000; ab75200, Abcam, Inc.), Cyp19a1 (1/1000; ab124776, Abcam, Inc.), and Cyp11a1 (1/500; #14217, CST, Inc., Danvers, MA, USA) were used to detect the levels of these proteins in granulosa cells.”
Point 8: Lines 132-134: This part is also found in lines 94-96.
Response 8: We have deleted this content, thank you very much for so careful reading.
Point 9: M&M: There is no description on the sampling procedures of sera and ovaries used for obtaining the results in lines 146-160. Therefore, lines 149-150, “The 8-wee-old….(n=6 per group), should be moved to M&M.
Response 9: We have moved“The 8-wee-old….(n=6 per group)” to the end of2.1. Animals, welfare assurance, and management (M&M.).
Point 10: Lines 162-163: “a series of …. and 10 ng/mL)” needs to be re-written.
Response 10: We rewrote this sentence with “leptin with different concentration (0, 0.1, 0.5, 1, 5, and 10 ng/mL)”
Point 11: Lines 162-169: There is no reasonable explanation/discussion on the quadratic response of the expression of cyc19a1, cyp11a1 and fshr genes to the leptin in any sections.
Response 11: Sorry, we changed the wrong expression with “leptin showed effects on the expressions of Cyp19a1, Cyp11a1, and Fshr genes”.
Point 12: Line 173: Is “RIA” correct? The authors describes “ELISA” in line 137.
Response 12: We corrected “RIA” with “ELISA”.
Point 13: Line 174-175: Check these sentences. The figure legends for “C and D” are wrong.
Response 13: We have corrected the figure legends, thank you.
Point 14: Line 180-181: Leptin did not stimulate Fshr gene expression, thus this sentence should be re-written;
Response 14: we have re-written this part as “To investigate the action of leptin on the expression of steroidogenesis genes, varied leptin doses (0, 0.1, 0.5, 1, 5, and 10 ng/mL) were added into cell culture media for 24 h and the expression levels of the E2 synthesis-related genes of Cyp19a1, Cyp11a1, and Fshr were determined. The observed changes of gene expression with changes in the leptin concentration series were different for different genes (Fig. 1). For the Cyp19a1 gene, the maximum expression appeared at 0.5-ng/mL leptin (p<0.05), and the minimum expression was at 10-ng/mL leptin (p<0.05, Fig. 1A). The expression of Cyp11a1was higher (p<0.05, Fig. 1B) at 0.1, 0.5, and 1-ng/mL leptin concentrations compared to the levels at the other leptin concentrations. Fshr expression was similar at 0, 0.1, 0.5, and 1 ng/mL leptin but was decreased at 5 and 10 ng/mL leptin (p<0.05, Fig. 1C).”
Point 15: Line 189: “Cyp9a1” should be “Cyp19a1”.
Response 15: We have corrected“Cyp9a1”with “Cyp19a1”.
Point16: Lines 196-197: A-D are not marked in Fig.3. This reviewer cannot imagine that cyp11a1 mRNA levels did not change by Lepr siRNA treatment from Figure 3. Please carry out the statistical analysis using the data for the potential Fig.3B. The authors stated they did not detect the protein of cyp11a1 but there are statement of cyp11a1 in the potential Fig. 3C and D. This part is so complicated to be understood.
Response 16: Thank you for finding my mistakes. I have corrected all of them. It is Fshr but not Cyp11a1 which was not changed by Lepron mRNA level. Therefore, we detected the protein level of Cyp11a1 but not Fshr. I have added the A-D on the top of every figure and changed the sequence of figures.
Point 17: There are possibly other careless and grammatical mistakes through the manuscript. Please check it out, again.

Round 2
Reviewer 2 Report
The authors carefully addressed my comments, thus the current version of this manuscript would be suitable for publication.